# AFOs Improve Stride Length and Gait Velocity but Not Motor Function for Most with Mild Cerebral Palsy

**DOI:** 10.3390/s23020569

**Published:** 2023-01-04

**Authors:** Hank White, Brian Barney, Sam Augsburger, Eric Miller, Henry Iwinski

**Affiliations:** 1Shriners Children’s Lexington, 110 Conn Terrace, Lexington, KY 40508, USA; saugsburger@shrinenet.org (S.A.); hiwinski@shrinenet.org (H.I.); 2Division of Physical Therapy, College of Health Sciences, University of Kentucky, 900 South Limestone, Lexington, KY 40536, USA; brian.e.barney@gmail.com; 3Pediatric Orthotic & Prosthetic Services—Midwest, 110 Conn Terrace, Lexington, KY 40508, USA; elmiller@shrinenet.org; 4Department of Orthopaedic Surgery, University of Kentucky, 740 S Limestone St., Room K423, Lexington, KY 40536, USA

**Keywords:** ankle–foot orthosis, motor function, minimum clinically important difference, over ground walking, gait analysis, cerebral palsy

## Abstract

Ankle–foot orthoses (AFOs) are prescribed to children with cerebral palsy (CP) in hopes of improving their gait and gross motor activities. The purpose of this retrospective study was to examine if clinically significant changes in gross motor function occur with the use of AFOs in children and adolescents diagnosed with CP (Gross Motor Function Classification System levels I and II). Data from 124 clinical assessments were analyzed. Based on minimum clinically important difference (MCID), 77% of subjects demonstrated an increase in stride length, 45% of subjects demonstrated an increase in walking velocity, and 30% demonstrated a decrease in cadence. Additionally, 27% of the subjects demonstrated increase in gait deviation index (GDI). Deterioration in gait was evident by decreases in walking speed (5% of subjects), increases in cadence (11% of subjects), and 15% of subjects demonstrated decreases in gait deviation index. Twenty-two percent of subjects demonstrated no change in stride lengths and one participant demonstrated a decrease in stride length. However, AFOs improved Gross Motor Function Measure (GMFM) scores for a minority (10%) of children with mild CP (GMFCS level I and II), with 82–85% of subjects demonstrating no change in GMFM scores and 5–7% demonstrating decrease in GMFM scores.

## 1. Introduction

Cerebral palsy (CP) describes permanent neuromuscular developmental disorders that occur because of disturbances in the fetal or infant brain [1]. The most common impairments seen in children with CP are decreases in motor control and presence of muscle spasticity [2]. These impairments may result in decreased ability to walk, perform transfers, and hinder standing balance. Limitations in ability to perform these activities can result in restricted abilities to participate in activities at home, school, and community with family and friends [3]. A variety of interventions (surgery, botulinum toxin injections, therapy, use of orthoses or assistive devices) can be used to decrease the severity of these impairments to improve gait and gross mobility [4]. Specifically, AFOs are often prescribed to facilitate functional activities (standing, walking and running) [5].

Ankle–foot orthoses (AFOs) are commonly used with children with spastic diplegia because literature reports AFOs improve temporal-spatial gait parameters and may indirectly effect knee and hip sagittal plane motions [6]. Systematic reviews of AFOs during walking report increased stride length and increased walking velocity compared to walking without AFOs [6,7,8].

Gait Deviation Index (GDI) and Gillette Gait Index (GGI) are indices that have been used to quantify overall kinematic profiles of gait quality. Further, GDI has been shown to be more sensitive to intervention than the GGI [9]. Some studies did not find a statistically significant difference in Gillette Gait Index (GGI) and Gait Deviation Index (GDI) scores between subjects in the barefoot and AFO conditions [10,11]. However, a study by Ries, Novacheck and Schwartz (2015) (with a larger number of subjects) found a statistical improvement in GDI, with 27 percent of the subjects having a clinically significant improvement [12]. However, the effects of AFOs on global measures of gait quality is less promising. Systematic reviews report mixed results for changes in gait quality for children with CP diplegia when wearing AFOs [6,7,8].

Gross motor skill activities, including standing, walking, running and jumping, are often impaired in children diagnosed with CP. AFOs are thought to improve these activities. However, the evidence of benefit for AFO use is less clear when assessing gross motor function for children diagnosed with CP. A recent meta-analysis of the literature reported four studies with 188 participants previously reported the effects of AFOs on GMFM. The pooled analysis revealed the effects of AFOs demonstrated small improvements sections D and E of GMFM with standardized mean differences (SMD) of 30 for sections D and 24 for sections E of the GMFM [5]. Of these four studies, some reported significant changes in GMFM scores while others reported no significant changes in GMFM scores when comparing barefoot to braced assessments [5].

The Gross Motor Function Measure (GMFM) is a criterion-referenced, valid, objective, reliable and responsive measure of activities for children diagnosed with CP [13]. The GMFM measures the capacity of children with CP to perform gross motor activities [14]. A systematic review reports both versions of the GMFM (GMFM-88, GMFM-66) have valid responsiveness for children and adolescents with CP [14].

These reviews, like most research studies, focus on comparing the average change in gait quality and GMFM scores based on conventional hypothesis tests assessing if statistically significant changes occurred beyond changes due to chance alone [15,16,17]. Obviously, assessing the average response of a group of subjects is very important. However, it is also well recognized that a small sample size can have insufficient power to demonstrate statistical significance and a large sample size can result in statistically significant changes that may not be clinically relevant [15,16,17]. Additionally, the response to a given intervention for a group of individuals will not be the same for each individual within the group.

One solution to this problem (an individual’s response being masked by the group response to an intervention) is to calculate the minimum clinically important difference (MCID) for a continuous measurement of interest. MCID is the “the magnitude of change required for an observable difference in function” and can be quantified using effect sizes (based on the distribution of variability of responses) [17,18]. MCID values have been established for the GDI [12], for the parameters of walking velocity, cadence, stride length, and Sections D (standing) and E (walking, running, and jumping) of the GMFM in children with CP [18]. By using the MCID of the GDI and GMFM the proportion of subjects demonstrating a clinically meaningful change can be reported. Additionally, characteristics of those subjects demonstrating meaningful changes compared to those not demonstrating a clinically meaningful change can be described [15,16,17].

Therefore, the primary purpose of this study is to determine if clinically prescribed hinged or solid ankle–foot orthoses in children with CP GMFCS levels I and II resulted in statistically and clinically significant improvements in gross motor function and gait quality based on based on MCID values and not just the mean changes in GDI or GMFM scores. We hypothesized that clinically and statistically significant increases in stride length, walking velocity, and GMFM scores occur in most subjects when wearing AFOs.

## 2. Materials and Methods

### 2.1. Recruitment Procedure

We performed an Institutional-Review-Board-approved, retrospective chart review of patients diagnosed with spastic diplegic CP. The data were obtained from an existing clinical database of children who underwent a three-dimensional motion analysis study, during the years of 1996 through and including 2016.

The inclusion criteria were patients who: (1) were referred to the Motion Analysis Center for a three-dimensional gait study as part of their plan of care, (2) were receiving their first gait evaluation, (3) had a primary diagnosis of cerebral palsy spastic diplegia, (4) were wearing the same AFO design on both legs at the time of the evaluation, (5) had not undergone any orthopedic surgeries in the past year, (6) had not undergone any Botox injections in the past six months, (7) had a documented diagnosis of cerebral palsy with gross motor functional classification system (GMFCS) levels I and II and (8) had a reported score of sections D and E of the Gross Motor Function Measure. All patients included in the study demonstrated at least neutral passive ankle dorsiflexion with the knee extended.

### 2.2. Characterization of Gait through Objective Parameters

More than one physical therapist performed the clinical three-dimensional gait evaluations. However, for each evaluation the same therapist performed all tasks. The patients performed two walking conditions (barefoot and with current AFOs and shoes) during a single visit. There was no control for sequence of conditions (barefoot or braced walking) for data collection. As part of quality assurance protocols at our facility, every six months the physical therapists collect kinematic and kinetic data with an able-bodied volunteer. Each assessment must demonstrate less than 5 degrees differences in the hip, knee and ankle sagittal plane data between the therapists’ assessments. Participants walked at their self-selected walking speed along a 10 m walkway wearing their typical clothes and shoes. A minimum of three strides and a minimum of two walking trails were collected and averaged. The first couple of strides and last couple of strides were excluded, as these could be transient steps of acceleration or deceleration [19].

Data were collected at 240 Hz using a Motion Analysis System with twelve Eagle digital cameras and Cortex (Version 5.50179) software (Motion Analysis Corporation, Santa Rosa, CA, USA). For both walking conditions (barefoot and with AFOs and shoes) participants wore thirty-four (34) surface reflective markers (Cleveland Clinic marker set) according to standardized clinical procedures. Using imbedded coordinate systems and Euler rotations, angles describing the rotation of one segment relative to an adjacent segment are calculated [20]. Each segment is free to translate and rotate independently of other body segments with a six degree of freedom model [20].

A gait cycle is defined from initial contact to the next ipsilateral initial contact [19,21]. By using definitive events (heel strike and toe off), the gait cycle is normalized to 100% and subdivided into phases (stance and swing). This normalization allows for comparing one walking condition to another.

Normative gait values for comparisons were obtained from our facility’s normative database. This was previously collected data of typically developing children between the ages of 4–21 years at our facility. Temporal-spatial and kinematic data are dependent on the events of foot strike and foot off for each walking trial and are represented as a percentage of the full gait cycle (100%) [22]. Walking velocity, cadence and stride lengths were reported as a percentage of typically developing age matched values. Participant’s gait patterns were reported as mean gait deviation indices. The gait deviation index (GDI) is a single value of a gait pattern using a Fourier transformation, which can take any waveform and convert it to a single number. The GDI uses gait kinematic data of the pelvic angles in all three planes, hip angles in all three planes, sagittal plane knee angles, sagittal plane ankle angles, and foot progression angles [23]. A typically developing child’s GDI is 100. A 10-point decrease in GDI score represents one standard deviation from the mean typically developed child’s gait pattern. The same process used to calculate GDI can be used to develop other indices based on different kinematic data, i.e., sagittal plane hip score [12]. Recently, a study calculated a knee-specific GDI using frontal and sagittal knee kinematics and joint moments using the same methods to calculate GDI [24]. The knee-specific GDI identified deviations in subjects with knee osteoarthritis compared to adults without knee osteoarthritis, while GDI scores alone did not show differences [24]. Therefore, we used the same process to calculate GDI to generate a single value representing sagittal plane hip (GDIh), knee (GDIk) and ankle (GDIa) kinematic data.

### 2.3. Characterization of Gross Mobility Gait through Objective Parameters

The therapist performing the gait study classified the patient’s gross mobility using the Gross Motor Function Classification System (GMFCS). The GMFCS classifies the motor involvement of children diagnosed with cerebral palsy based on self-initiated movement [25]. The classification system has an emphasis on determining the child’s present abilities and limitations in motor function. There are five levels of motor control in the GMFCS with Level I demonstrating the fewest impairments and Level V the most.

At our facility, all therapists participated in a GMFM training course and passed the training examination prior to assessing patients. For clinical gait evaluations at our facility sections D (standing) and E (walking running and jumping) of the Gross Motor Functional Measure (GMFM) were performed during the same visit as the gait analysis evaluation. The GMFM is a standardized observational clinical test that measures change in gross motor function in children with cerebral palsy [18]. The data reported is in percentages of tasks completed. A typically developing 5-year-old child should score a 100% for sections D and E [18].

### 2.4. Characterization of Ankle–Foot Orthosis

All patients wore custom made hinged or solid AFOs at the time of the gait study. Patients wearing ground reaction AFOs and supra-malleolar orthosis were excluded. Because this is a retrospective study spanning a twenty-year period a wide variety of materials were used to make the solid AFOs including copolymer plastic polypropylene and polyethylene materials of varying thicknesses (3.2–4.8 mm) with anterior trim lines anterior to each malleolus. Type of plastic, thickness of plastic, and footplate lengths were based on clinicians’ assessment of patient’s weight and activity level. The hinged AFOs were made similar to solid AFOs with modification using a Tamarack hinge joint at the level of the medial and lateral malleoli with free dorsiflexion movement and 90-degree plantar flexor stop. For both hinged and solid AFOs, hook and loop straps were positioned at the front of the ankle and anterior tibia. Proximally, the AFOs were 3–5 cm below the popliteal fossa.

### 2.5. Minimum Clinically Important Difference (MCID)

Minimum Clinically Important Difference (MCID) used in this study for GMFM scores, walking velocity, cadence and stride lengths, are based on the MCID values of a large (0.9) effect size established by Oeffinger et al. [18]. The MCID were calculated from two assessments (time between assessment: 1.4 years) without surgical interventions from over 130 participants [18]. The MCID for GDI changes has been reported to be 5 points [12] (Table 1).

### 2.6. Statistical Analysis

Statistical analyses were performed using SPSS version 24 (IBM Inc., Chicago, IL, USA). General linear models repeated measures analysis of variance (GLM-ANOVA), with GMFCS level and brace type (hinged vs. solid) as covariates, were performed to assess mean changes between braced and barefoot walking and GMFM scores. GLM-ANOVA statistical tests assess a variable’s mean changes and for an interaction between covariates, which indicates the groups are changing differently. Mann–Whitney U Test (nonparametric *t*-tests) were used to assess differences between those participants that demonstrated a significant increase or decrease in GMFM scores based on MCID values. Chi-square tests were used to compare frequencies of ordinal (GMFCS levels) or nominal variables (sex, AFO type) between those participants that demonstrated a significant increase or significant decrease in GMFM scores based on MCID values. Effect size (Cohen’s d) between barefoot and braced conditions for GMFM scores and gait parameters were also calculated.

## 3. Results

An initial query identified 546 potential patients. Based on inclusion criteria 124 children with a diagnosis of spastic diplegic CP GMFCS levels I and II (from here on referred to as mild CP) were included in this study. Subjects’ mean age was 8.8 ± 3.3 (range 4–18) years at the time of the study. Descriptive characteristics of the subjects are in Table 2.

The results of subjects were categorized by significant increase, no change, or significant decrease (based on MCID values) in gait quality and GMFM scores, comparing AFO use and barefoot (Table 3).

General linear models repeated measures analysis of variance (GLM-ANOVA), with GMFCS level and brace type (hinged vs. solid) as covariates, were performed to assess mean changes between braced and barefoot walking and GMFM scores. Statistically significant increases within subjects were observed from the barefoot to the AFO condition for walking velocity (F [1, 120] = 57.8, *p* < 0.001), stride length (F [1, 120] = 158.3, *p* < 0.0001), and GDI (F [1, 120] = 5.1, *p* = 0.025), but not for walking cadence or GMFM scores. A statistically significant increase in the percentage of the gait cycle spent in swing phase (F [1, 120] = 11.1 *p* = 0.001) and a significant decrease for stance phase (F [1, 120] = 11.1, *p* = 0.001) were also observed. There were no significant interactions for brace type or GMFCS levels for temporal-spatial data; however, there were interactions for GDI score. For changes in GDI score there was a significant interaction between AFO type and GMFCS levels (F [1, 120] = 9.1, *p* = 0.003). There was also a significant difference between GMFCS levels (F [1, 120] = 5.10, *p* = 0.026) with GMFCS level I demonstrating larger GDI than level II for both walking conditions. Subjects classified as GMFCS level I, who wore hinged AFOs, demonstrated a larger increase in GDI scores, while those wearing solid AFOs demonstrated small decrease in GDI scores. Subjects classified as GMFCS level II wearing solid AFOs demonstrated an increase in mean GDI scores; those wearing hinged AFOs demonstrate no change in GDI score (Table 4).

For joint specific GDI scores, there were not significant changes in GHIh and GDIa scores. However, there were significant changes in GDIk scores: right knee (F [1, 120] = 7.66, *p* = 0.007) left knee (F [1, 120] = 7.27, *p* = 0.008) (Table 3). There were no significant interactions for brace type or GMFCS levels for joint specific GDI scores.

Clinically significant differences in gait within subjects from barefoot to AFO included an increase in stride length in the majority (77.4%) of subjects, an increase in walking velocity in nearly half of all subjects (45.2%), an increase in GDI score in just over a quarter of subjects (27.4%), and a decrease in walking cadence in 30% of the subjects.

However, only 10.5% of the subjects had an MCID-level increase in Sections D and E of the GMFM when comparing barefoot to AFOs. Based on MCID values; subjects who experienced a meaningful increase in section D scores had significantly lower barefoot Section D GMFM and GDI scores compared to those with a significant decreased (*p* = 0.025 and 0.002, respectively) (Table 5). Additionally, there was a statistically significant difference in the percentage of subjects who saw an MCID-level increase in GMFM Section D scores based on their GMFCS level (*p* = 0.043).

Large effect sizes between barefoot and braced conditions for stride lengths and walking velocity were demonstrated. GMFM scores and other gait parameters demonstrated small effect sizes (Table 6).

## 4. Discussion

The primary purpose of this study was to determine if clinically prescribed hinged or solid ankle–foot orthoses in children with mild CP resulted in clinically and statistically significant improvements in gross motor function and gait quality based on MCID values. We hypothesized that clinically and statistically significant increases in stride length, walking velocity, and GMFM scores occur in most subjects when wearing AFOs.

The mean stride length and velocity demonstrated clinical and statistically significant increases when comparing walking trials with AFOs to barefoot walking trials. Percent of the gait cycle in stance and swing changed less than 1%. These results were not presented because change is not clinically significant. However, mean scores of sections D and E of GMFM did not demonstrate statistically significant increases.

Because of the heterogeneous changes of the braced conditions reported we believe it is beneficial to also assess subjects based on MCID values and not just the mean change for all participants. In terms of clinically important differences in gait parameters based on MCID values, 77% of subjects demonstrated increased stride length. These results are very similar to the Ries et al. study, which reported that 76% improved their step length [12]. Additionally, only 45% of our subjects demonstrated significant increase in walking velocity which is similar to Ries et al. results (48% of their subjects improved their walking speed) [12]. One of the reasons for small difference in results (between Ries et al. [12] and our study) could be due to how velocity was reported. For our study, walking velocity is reported at a percentage of mean value for age-matched typically developing children. Reis et al. [12] reported walking velocity as nondimensionalized value by dividing walking speed by square root of acceleration due to gravity multiplied by leg length. These differences in calculating walking velocity may be one reason why inconsistent results have been reported for the magnitude of increase in walking speed reported in systematic reviews. Three systematic reviews of the literature report similar findings of increases in step or stride lengths, but mixed changes in walking speed and cadence for children with spastic diplegia CP [6,7,8].

Deterioration in gait is not typically reported in the literature for AFO use for children with CP diplegia. In our study, decreases in walking speed (5% subjects), increases in cadence (11% subjects), and 15% of subjects demonstrated decreases in gait deviation index. Twenty-two percent of subjects demonstrated no change in stride lengths and one participant demonstrated a decrease in stride length.

Our results also reveal a statistically significant improvement in GDI from barefoot to AFO and a clinically important increase in GDI scores in 27% of subjects, which is the same percentage of subjects that Ries et al. reported [12]. One reason for this agreement is both studies used the same method to calculate GDI. The GDI is calculated by using three-dimensional gait kinematic data from the pelvis, hip, knee, ankle and foot and comparing it to normal kinematics [23]. It is possible for AFO use to have greater effects on the kinematics of some joints more than other joints, and these effects may not be large enough to increase total GDI values by the MCID. Ries et al. reported the GVS of the ankle and knee, with 31% of subjects demonstrated a significant increase in ankle GVS and 10% demonstrated an increase in knee GVS [12]. In our study, joint specific GDI scores were calculated for the hip, knee and ankle using the same methodology to calculate GDI. Our results demonstrated no significant change for GDIh or GDIa scores and small increases in GDIk scores (Table 3). The nonsignifcant change in GDIh scores were expected. The statistically significant, but likely not clinically significant, changes in GDIk scores reported are similar to Ries et al. findings and reviews of the literature [6,7,8,12]. Our nonsignifcant changes in GDIa scores were not expected. We propose this could be due to the majority of the subjects wore hinged AFOs. Reviews of the literature report similar finding of mixed effects of AFOs on overall gait quality and joint specific changes [6,7,8].

To date, we are not aware of any other studies that have used MCID values to assess the effect of AFO use on GMFM scores in children with CP. In our study, on average, nonsignifcant (1 unit) increases in GMFM scores were demonstrated. AFOs improved GMFM scores for a small minority of children with CP (GMFCS level I and II). Based on MCID values, over 80% of the subjects demonstrated no change in GMFM scores. Only 10% of subjects demonstrates a clinically significant improvement in GMFM scores based on MCID values and 5–7% of subjects demonstrated decreases in GMFM scores. When performing the GMFM barefoot, 8% of subjects scored the maximum for section D and 4% of subjects scored the maximum for section E, indicating a small ceiling effect.

When comparing those subjects who demonstrated a significant increase in section E of the GMFM to those with a significant decrease, there were no statistical differences (based on Mann–Whitney U *t*-test) between the two groups. Therefore, we were not able to identify any differences between the two groups.

When comparing those subjects who demonstrated a significant increase in section D of the GMFM to those with a significant decrease (based on Mann–Whitney U T-test) there were significant differences between the two groups barefoot section D GMFM and barefoot GDI. Those subjects who demonstrated a significant increase in section D of GMFM had lower barefoot GMFM scores and lower GDI scores (indicating more abnormal gait pattern) compared to those subjects that did not demonstrate a significant increase in section D. Therefore, those with more impairments in gross mobility and gait quality benefited more from AFOs. One reason that section D of GMFM demonstrated differences could be due to the activities performed in section D, which included transitional movements (sit to stand, getting up from the floor, and half-kneeling) while section E of GMFM does not.

Four previous studies reported statistically significant differences in subjects’ GMFM scores with and without AFOs [26,27,28,29]. For these studies and ours, the standard deviation for sections D and E of GMFM when wearing AFOs ranged from 10–20%. Large standard deviation could indicate increases and decrease in scores [29]. However, to date, only mean and standard deviations have been reported for changes in GMFM scores [26,27,28,29]. To our knowledge, our current study is the first to report changes in mean GMFM scores and changes in GMFM scores based on MCID. By reporting the changes in GMFM scores based on MCID, we demonstrate only 10% subjects show a significant improvement in GMFM scores.

Our results are consistent with a recent review of the literature reported four studies of 188 participants pooled analysis revealed the effects of AFOs demonstrated small improvements sections D and E of GMFM with standardized mean differences (SMD) of 30 for sections D and 24 for sections E of the GMFM [5]. The standardized mean difference is used in meta-analysis and is mean difference between two groups divided by their respective standard deviation, also known as Cohen’s d [30]. Therefore, we calculated the effect size of the GMFM and gait measures. The effect size for sections D was 0.14 and section E was 0.17. These values are smaller than the standardized mean differences (SMD) of 30 for sections D and 24 for sections E of the GMFM reported from meta-analysis of effects of AFOs on gross motor function [5]. These differences in effect size could be due in part to the subjects in our study were less involved (based on GMFCS levels) and older in age. Because small effect size of changes in GMFM scores with AFO, the mean changes are not indicative of individual responses for all subjects. Therefore, we believe it is also useful to use MCID values to assess the percent of individuals demonstrating a clinically meaningful change.

### Limitations

The limitations of this study are primarily due to its retrospective nature, resulting in the lack of information on the specifics of the ankle–foot orthoses used by the subjects. Some subjects had previously received orthopedic surgery or other interventions, such as Botulinum toxin injections, while others had not. However, we excluded subjects who had interventions more recently (Botulinum toxin injections in past 6 months, surgery in past year). Additionally, because our participants were GMFCS levels I and II, results of this study cannot be generalized to wide population of children with CP. The next limitation is that an explanation of the physicians and orthoptists’ rationale for type of brace for the patient was not available. Unfortunately, this information could not obtain from a retrospective chart review. Therefore, we do not have any information regarding if AFOs were prescribed appropriately. Perhaps subjects with inappropriately prescribed AFOs had worse outcomes. Different types and thickness of different plastics were used to fabricate the AFOs. Therefore, the stiffness of these braces based on materials used could not be assessed. Additionally, the type of shoes worn by patients was not controlled. Overall, the lack of information on the AFOs used and the heterogeneity of our subject population is consistent with other retrospective studies. No optimization of the shank-to-vertical angle (SVA) of the AFO footwear combination or AFOFC is a major limitation of this study as studies have shown that tuning can contribute to improved gait outcomes [31]. Additionally, we did not measure the AFOFC at mid-stance to determine whether their braces were tuned correctly. However, each AFO was aligned based on orthotists’ clinical visual observation. Future studies could assess the AFOFC at mid-stance from these data and could potentially identify those patients that demonstrated improvements in gait and GMFM had optimal AFOFC, compared to those that demonstrated no improvements or worsening gait and GMFM score.

## 5. Conclusions

This study confirms previous reports of the effects of AFO use on gait parameters and gross mobility. Additionally, new information based on MCID is presented. A majority (77%) of subjects demonstrated a statistically and clinically significant increase in stride lengths, resulting in almost half (45%) of subjects demonstrate improvements in walking speed based on MCID values. Only 27% demonstrated an increase in walking quality. Joint specific GDI values were not impacted at the hip or ankle; however, knee GDI scores showed improvements. AFOs improved GMFM scores (based on MCID values) for 10% of children with mild CP (GMFCS level I and II), with 82–85% of subjects demonstrating no change in GMFM scores and 5–7% demonstrating decreases in GMFM scores. When considering AFOs to improve a patient’s gross mobility or walking, an assessment using GMFM, or other outcome tools, should be performed to demonstrate the AFOs benefit the individual.

## Figures and Tables

**Table 1 sensors-23-00569-t001:** Minimum clinically important difference (MCID) for GMFCS Levels I and II from the literature [12,18].

GMFM and Gait Quality	GMFCS Level I MCID Value	GMFCS Level II MCID Value
GMFM Section D score	3.8	5.3
GMFM Section E score	6.5	4.5
Walking Velocity ^#^	13.9	10.9
Walking Cadence ^#^	9.5	12.2
Walking Stride Length ^#^	6.7	6.3
GDI score	5	5

^#^ Values for these measures reported as percentage of mean values for age-matched typically developing children.

**Table 2 sensors-23-00569-t002:** Demographics descriptive statistics, *n* and percentage of each group.

Demographics	*n* (%)
Sex	
Male	79 (63.7%)
Female	45 (36.3%)
AFO type	
Solid AFO	35 (28.2%)
Hinged AFO	89 (71.8%)
GMFCS Level	
Level I	39 (31.5%)
Level II	85 (68.5%)

**Table 3 sensors-23-00569-t003:** GMFM and gait quality changes for AFO use with percentages of subjects with clinically significant improvements.

GMFM and Gait Quality	Mean BF Value	Mean AFO Value	*p* Value	>MCID (%)	No Change—MCID (%)	<MCID (%)
GMFM Section D	86.8 ± 8.5	87.5 ± 8.2	NS	10.5%	82.3%	7.3%
GMFM Section E	75.7 ± 16.6	76.4 ± 16.5	NS	10.5%	84.7%	4.8%
Walking velocity ^#^	81.0 ± 21.0	91.0 ± 20.0	<0.001	45.2%	50.0%	4.8%
Walking cadence ^#^	102.0 ± 17.1	100.0 ± 13.2	NS	11.3%	58.9%	29.8%
Stride length ^#^	78.0 ± 16.2	92.0 ± 15.3	<0.0001	77.4%	21.8%	0.8%
GDI	63.2 ± 10.0	64.8 ± 10.6	0.025	27.4%	57.3%	15.3%
R Hip GDI	84.8± 9.8	84.7 ± 9.4	NS	N/A	N/A	N/A
L Hip GDI	83.5 ± 10.1	83.2 ± 9.9	NS	N/A	N/A	N/A
R Knee GDI	69.2 ± 8.1	70.3 ±9.1	0.015	N/A	N/A	N/A
L Knee GDI	68.5 ± 7.9	69.7 ± 9.1	0.017	N/A	N/A	N/A
R Ankle GDI	74.9 ± 13.9	76.4 ± 8.7	NS	N/A	N/A	N/A
L Ankle GDI	74.9 ± 14.4	75.7 ± 9.1	NS	N/A	N/A	N/A

^#^ Values for these measures reported as percentage of mean values for age-matched typically developing children. R = right limb; L = left limb; NS = non-significant (*p* value ≥ 0.05)

**Table 4 sensors-23-00569-t004:** Descriptive statistics changes in GDI scores.

Demographics	Mean ± Standard Deviation BF Value	Mean ± Standard Deviation AFO Value
GMFCS Level I		
All subjects	75.2 ± 12.8	76.4 ± 6.4
Hinged AFO	72.7 ± 13.5	76.8 ± 7.1
Solid AFO	81.4 ± 8.1	75.5 ± 7.2
GMFCS Level II		
All subjects	74.8 ± 14.5	76.5 ± 9.6
Hinged AFO	75.6 ± 14.9	75.8 ± 9.4
Solid AFO	72.9 ± 13.6	78.2 ± 10.0

**Table 5 sensors-23-00569-t005:** Mean barefoot values for gait quality and GMFM scores based on significant changes in Sections D and E of GMFM with use of AFO.

GMFM and Gait Quality	Significant Decrease in Section D of GMFM Based on MCID	Significant Increase in Section D of GMFM Based on MCID	*p* Value
GMFM Section D	88.0 ± 6.5	79.2 ± 8.8	0.025
GMFM Section E	79.4 ± 15.3	68.5 ± 16.8	0.126
Walking velocity ^#^	69.2 ± 17.4	72.3 ± 14.7	0.695
Walking cadence ^#^	98.6 ± 16.7	96.3 ± 11.1	0.896
Walking Stride length ^#^	71.2 ± 16.5	75.0 ± 10.6	0.601
GDI score	70.7 ± 7.1	59.2 ± 7.1	0.002
**GMFM and Gait Quality**	**Significant Decrease in Section E of GMFM Based on MCID**	**Significant Increase in Section E of GMFM Based on MCID**	** *p*** **Value**
GMFM Section D	86.0 ± 6.2	81.9 ± 8.9	0.269
GMFM Section E	73.2 ± 12.0	76.8 ± 16.9	0.397
Walking velocity ^#^	89.6 ± 22.0	84.0 ± 23.1	0.624
Walking cadence ^#^	111.6 ± 15.6	103.2 ± 15.6	0.210
Walking Stride length ^#^	79.7 ± 15.1	80.6 ± 13.2	0.907
GDI score	61.0 ± 14.3	61.2 ± 8.7	0.835

^#^ Values for these measures reported as percentage of mean values for age-matched typically developing children.

**Table 6 sensors-23-00569-t006:** Effect sizes (Cohen’s d) of GMFM scores and gait parameters.

GMFM and Gait Quality	Cohen’s d
GMFM Section D	0.14
GMFM Section E	0.17
Walking velocity ^#^	0.76
Walking cadence ^#^	0.18
Stride length ^#^	1.29
GDI	0.30
R Hip GDI	0.04
L Hip GDI	0.04
R Knee GDI	0.22
L Knee GDI	0.22
Right Ankle GDI	0.09
Left Ankle GDI	0.05

^#^ Values for these measures reported as percentage of mean values for age-matched typically developing children. R = right limb; L = left limb.

## Data Availability

Not applicable.

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
