# Peer review of "AFOs Improve Stride Length and Gait Velocity but Not Motor Function for Most with Mild Cerebral Palsy"

_sensors, 2023, doi:10.3390/s23020569_

Round 1
Reviewer 1 Report
Comments and Suggestions for Authors
Thank you for the opportunity to review this manuscript. The authors retrospectively reviewed gait and GMFM sections D & E results from 124 children with CP, GMFCS I and II, during barefoot gait and gait with shoes and AFOs, to clarify effects of AFOs on gross motor skills. I agree this is a question of relevance to clinicians who work with this population of children. This type of research has a number of limitations that make this a daunting task. I appreciate the value of shifting focus to explore clinical, rather than statistical significance. However, I am not convinced that the study can achieve the intended aims. Major revisions may provide a better opportunity to judge the study’s potential contributions.
One significant concern is that the gait measure (GDI) lacks validity for the purpose of orthotic evaluation (Galli et al., 2016; Danino et al., 2015; Danino et al., 2016). The GVS (Baker et al) has been proposed as a more specific measure for this purpose. This issue has not been addressed in the paper at all. It is also difficult to follow the research questions, data analysis and results clearly. As a result, the discussion is confusing to read, and the results and implications appear very much overstated.
The discussion would benefit from substantial revision and reorganization to focus it on the results, specifically as they pertain to the research questions, and to avoid overstating the findings. Spelling and grammatical errors should be corrected.
Introduction
I would suggest updating the papers you have cited to summarize the current state of the evidence using papers with stronger levels of evidence and more current references. For example, Reference 4 is 30 years old, and presently there is a lack of convincing evidence that AFOs improve range of motion or function, or prevent deformity (e.g., Bowers and Ross, 2009) Similarly for lines 33-35, it would be advisable to cite meta analyses or systematic reviews, rather than papers with small sample sizes.
A focus of the introduction should be on the effects of AFOs on gross motor skill. More thorough examination of the background would be useful here. Why might AFOs improve gross motor skill? Specifically what improvements in gross motor skills have been reported? I would also suggest focusing the summary of effects on gross motor skills on higher quality evidence. For instance, the first papers mentioned 12 and 13 are small studies, which provide only very weak evidence that is not generalizable.
I suggest revising and reorganizing the introduction to justify the aim more clearly. By line 67, It is completely unclear what “the one previous study” is, and why it is worth exploring.
Methods
- Line 91: Do you have data for inter rater reliability for the assessment procedures? If not, this should be included as a limitation
- The 20-year time frame is likely to introduce additional variables. Why was there only data for 124 children over this large timeframe?
- Did you do a power analysis? How do you know the sample is large enough to detect a difference?
- To my knowledge a t test is a parametric test. What is a nonparametric t test?
- This section would benefit from description of the psychometric properties of the GMFM, for example, the validity of measuring a single section of the test, and both the interrater, and test re-test reliability.
- What ordinal and nominal variables were compared using Chi square tests? It would be helpful to clarify what variables were compared and by which tests.
- Specifically which MCIDs did you use for each variable? When looking at table 2 it would be helpful to know what the MCID for each variable was. This may help readers apply the findings.
A serious flaw of this paper is the lack of established validity and sensitivity of the GDI as capable of identifying change with an AFO. It is my understanding that global gait indices (i.e., the GDI, or GPS) have not been demonstrated to be specific enough to identify the effects of focussed orthotic intervention on gait parameters such as ankle kinematics that are most likely to be impacted by AFOs (e.g., Danino et al., 2015; Danino et al., 2016; Galli et al., 2016). Gait Variable Score (Baker et al) is likely more promising (Galli et al., 2016; Kane et al., 2020). This should be addressed in the introduction, methods, and discussion.
Results
Table 1 is confusing to look at for a number of reasons. For example, Sex is the header for all of the left column data, Male is in bold font, and there is a line between male and female. Please reorganize this table. GMFCS levels should be referred to by Roman numerals.
Line 173-176: what outcome measure are you referring to? (Also there is a typo “an outcome measures”)
Line 173-74: It is also confusing to say that ”subjects were grouped by” whether they had a change for a particular variable. This sounds like something you did to the participants, rather than their data/results. Please reword this to reflect the reporting of results.
Line 182-187: You report that there was a significant interaction between AFO type and GMFCS levels for change in GDI score. The supporting data should be reported in the tables. Was this difference clinically significant?
Table 2: results for GDI and stride length are very hard to read due to the spacing; values in the 3 right hand columns are also hard to read
Line 203- 208: Given that the effect on GMFM score was not significant, it is important to also report the % who experienced a negative effect.
Table 3 and lines 212-214 are unclear. There are typos in this sentence. What conditions are referred to?
Please be sure to make it clear which tests were done to produce each of the results. It would help to edit the analysis section as well, to ensure clarity. Specifically, how does the data in table 3 relate to your analysis? And why did you only seem to do this analysis for the data from section D of the GMFM?
Discussion:
-The stated purpose in the first sentence should match the one at the end of the introduction.
-line 228: walking ability is a broad term. I’m not sure that’s what you measured here. Gait quality may be more accurate
-line 231-234: as per my question above, this analysis needs more detail presented. Table 3 and that part for the results only describe this relationship for section D, but this paragraph says the findings were the same for section E. Please present the section E data/results as well. I have the same comment for lines 271-274. It seems like there is a lot of data missing from the results for these other variables.
-line 232: which outcomes specifically? This is misleading without specificity.
-In general, the significance and implications of results are overstated throughout. Please soften the strength of the language throughout the abstract, discussion, and conclusion.
For example, the final sentence of the first paragraph is not substantiated at all. This is a small group of children with heterogenous (undescribed) clinical presentations and orthotic designs. The goals of the AFOs are also unknown. There may be other clinical benefits to the children that you have not measured.
Therefore, if this paper is to be revised, I suggest that the discussion remain specific to the direct findings and acknowledge the numerous limitations and inability to generalize to the wider population of children with CP.
-line 245: to be clear, the changes in stance and swing time were around 1%, so not likely to be clinically meaningful. Please state this clearly.
-line 256: Did you demonstrate that velocity improved more for kids with greater impairments? I did not see this data presented.
- line 258: The final sentence of this paragraph is also confusing. It would be helpful to clearly state the benefits you’re trying to assert, to make a more effective argument. Why exactly would someone come to that erroneous conclusion? Do you mean that looking only at group averages may mislead one to think there was an effect for all kids with spastic diplegia? Is the point rather that within your group there was a variety of responses? This is one of the limitations of focussing on measures of central tendency with children with CP (who are a heterogenous population) (e.g., Damiano, 2014). I am wondering if the benefit of the MCID you’re trying to describe is the ability to judge the clinical significance of an individual’s response (e.g., the potential to draw more meaningful clinical conclusions from effect size data as compared to statistical significance).
-line 289 – this statement seems to contradict other statements you made (e.g., line 231-234)
-Line 275-305: this paragraph lacks focus, so it is difficult to appreciate what it is trying to say. It would help if this could be edited to focus on a couple of meaningful points that are justified by the results.
-line 326-328: I would suggest using a more current and accurate definition of tuning and biomechanical optimization. See Elaine Owen’s recent publications (e.g., 2019-2021 editor’s comments in JPO for instance) for a more accurate definition. The lack of tuning is definitely a limitation.
-Given the lack of evidence of responsiveness, validity, or specificity of the GDI for orthotic evaluation, it is advisable to provide a possible explanation for the observed results, and place the results in the context of the literature.
Conclusion:
line 343-345: as per comments above, I do not see that the data has been provided to substantiate this statement.
It is also unclear what types of impairments were associated with the improvements.
Abstract
Line 16: use of the term gait studies is potentially misleading. Please use terminology that clarifies that your examined clinical assessment results not research studies.
-Why not also mention the effect on cadence and step length?
-If you mention improvements in variables like GMFM D & E score, you should also mention what % demonstrated a deterioration in that variable. Otherwise, the presentation of your results appears biased and incomplete. (E.g., re: GMFM scores, the effect was not significant, so kids were statistically as likely to improve as to decline with the AFOs)
Reviewer 2 Report
Comments and Suggestions for Authors
well-written resubmission.
Round 2
Reviewer 1 Report
Comments and Suggestions for Authors
Thank you for the opportunity to review this revised manuscript. It describes a retrospective review of 124 children with CP, spastic diplegia, GMFCS I and II over a 20-year period. It aimed to determine whether the use of hinged or solid AFOs impacted gait quality (GDI, lower extremity joint-specific GDI, and spatio-temporal parameters) or gross motor skill (GMFM score).
I appreciate that the authors have made substantial revisions in attempt to address the concerns described in the initial review. Thank you for including missing data in this revision, and more fully explaining some of the key points that were omitted earlier. However, I continue to have substantial concerns about the quality of the manuscript, the meanginfulness of results, and the contribution it may make to the evidence if it was to be published. Therefore, unfortunately, I am not recommending publication. I have outlined my concerns below:
1. The quality of writing is poor in many places, and most paragraphs have at least one grammatical or technical error (e.g., Gross Motor Functional Classification System, and Gross Motor Functional Measure are not the correct names of these measures). I’m not sure if some of these errors were the result of hasty revisions, but it impacts my impresion of the manuscript quality. As well, some more minor points:
a. Some of these errors suggest a lack of understanding of the topic. E.g., Line 35: “increases” in spasticity do not characterize CP. Increases relative to what? The presence of spasticity may be more appropriate to mention. Also, what does it mean to “perform standing balance”?
b. I do not agree with the change to the title. It is too long. The question “do AFO’s really help children with CP” is far too vague, and is not answered by the study.
2. The introduction does not support the study effectively. I continue to have concerns about the validity of the GDI as an outcome for this study. You have not clearly described its ability to detect change for this purpose, or what is known about this. The studies you cite both demonstrate a lack of effect of AFOs on GDI. Hence, this does not demonstrate a compelling argument for the study.
3. It is not clear why effect size results are suddenly added into the discussion.
4. The AFO literature is generally of low quality, and higher quality studies are needed in order to advance the state of the evidence. It is not sufficient to say that the study has limitations that are similar to other studies. It is important to improve the quality of the research rather than continue to publish the same types of studies that were done in past decades. It is important to note that her that the validity of your conclusions are severely impacted by a number of factors. For instance, the AFOs were made over a span of 20 years, while materials and approaches have been changing substantially, the AFOs are completely without description, the goals of the AFOs were not known, the solid AFOs are not tuned. The lack of detail about participants (e.g., ankle dorsiflexion range of motion, gait patterns) makes it impossible to judge the appropriateness of the hinged or solid AFO types that were selected or draw conclusions about the effects of either. For example, see Ridgewell et al., 2010 for a review of reporting recommendations for AFO research.
So, even looking beyond the significant challenges posed by the absence of AFO descriptors, it is unclear to me how these findings would contribute to the literature. The primary results appear to be that the biggest improvements with the AFOs were seen in the spatio-temporal results. This is not a novel conclusion in the literature. The other conclusion you highlight is that the effects of AFOs on gross motor skills is unclear, and this is also already reported in the literature.
Unfortunately, given the lack of detail and heterogeneity of AFOs used in the study, it does not seem that the results advance the study or understanding of the effects of AFOs for children.
Author Response
We greatly appreciate all of the time and effort that this reviewer has provided to us. Please see attached pdf file with point-by-point response to reviewer's comments.

Round 3
Reviewer 1 Report
Comments and Suggestions for Authors
Thank you for the opportunity to re-review this manuscript. The authors have made substantial edits to improve the paper. If the paper was to be published, I would suggest further revisions to address the following outstanding concerns:
- There continue to be numerous errors in grammar, spelling, and syntax (e.g., line 40, 43, 66/67, 74, 103, 119-121, 138, 144, 156, 168, 189, 206-207, 340-341, 349, 356, 359, 380, 381, 403, 408, etc.) that interfere with the reader’s understanding. It seems that many of these errors were present in the previous revisions, but I highly suggest the authors make an effort to correct them, as this would improve credibility and understanding of the content. The discussion in particular, is hard to understand in many places, due to these errors.
Title, abstract, introduction
- I appreciate that you changed the title. However, to say that “AFOs improve gait” is too general. You report that AFOs may improve a certain aspect of gait. But as it is, the title implies that AFOs affect gait in general, which is misleading/ Please consider revising the title , as the term “gait” is too general to be accurate.
- The term “mild CP” used throughout the paper to describe the sample, is too subjective, and nonspecific. It would be more accurate and helpful to readers if you could describe the sample by using a more meaningful/specific term (e.g., refer to the GMFCS levels included).
- Line 59. The introduction here still does not make logical sense to me. You state that the effect of AFOs on gross motor skills is less clear, but then state AFOs are thought to improve gross motor activities. It would make more sense if you described the uncertainty or discrepant effects that have been reported in the literature.
- Line 93: you refer throughout the paper to the aim of the study as to explore changes in gait and GMFCS levels based on “changes in MCID values”. This implies that MCID values change, but they do not. Please clarify in all relevant places in the manuscript (including discussion)
-
Methods
- Please provide a reference for the validity of the GDIh, GDIk, and GDIa variables
- Line 157 – is this supposed to say GMFM training?
- Line 162- the GMFCS does not quantify motor control
- Line 195: how were MCID values determined? Please provide more information about the MCID in this section. MCID information is currently in the results section (e.g., line 208), but this is more appropriate for the methods section.
Results
- Table 3 is not your results. This would be better suited to methods/analysis. Currently it looks like this is your data, as the table is not even referenced, and it is located in the results section. Table should be referenced appropriately.
- Line 264-265: where is this data from? How was it calculated?
Discussion
-line 342: To say that those with lower barefoot scores and poorer gait quality “benefitted more from AFOs” is not specific enough. Please specify what aspect of their performance appeared to benefit.
- line 359-369: I appreciate your response about the rationale for the effect size calculations. The intent of my feedback was to ask that this information be placed – with appropriate detail and information - in the methods and results sections. It is not appropriate to introduce an analysis in the discussion.
Conclusions: are there clinical implications?
Author Response
Reviewer:
We greatly appreciate all of your comments and suggestions for this manuscript. We also would like to thank you for your patience in this process. Your original comment "some of these errors were the result of hasty revisions, but it impacts my impression of the manuscript quality."
We have addressed all of your comments and concerns to the best of our abilities.
Thank you again for all of your time.
